# ADCY7 mRNA Is a Novel Biomarker in HPV Infection and Cervical High-Grade Squamous Lesions or Higher

**DOI:** 10.3390/biomedicines11030868

**Published:** 2023-03-13

**Authors:** Lihua Chen, Lixiang Huang, Binhua Dong, Yu Gu, Wei Cang, Chen Li, Pengming Sun, Yang Xiang

**Affiliations:** 1Department of Obstetrics and Gynecology, Peking Union Medical College Hospital, Chinese Academy of Medical Sciences, Peking Union Medical College, Beijing 100730, China; 2National Clinical Research Center for Obstetric & Gynecologic Diseases, Peking Union Medical College Hospital, Chinese Academy of Medical Sciences, Peking Union Medical College, Beijing 100006, China; 3Laboratory of Gynecologic Oncology, Fujian Maternity and Child Health Hospital, College of Clinical Medicine for Obstetrics & Gynecology and Pediatrics, Fujian Medical University, Fuzhou 350001, China; 4Fujian Key Laboratory of Women and Children’s Critical Diseases Research, Fujian Maternity and Child Health Hospital (Fujian Women and Children’s Hospital), Fuzhou 350001, China; 5Fujian Clinical Research Center for Gynecological Oncology, Fujian Maternity and Child Health Hospital (Fujian Obstetrics and Gynecology Hospital), Fuzhou 350001, China

**Keywords:** ADCY7, cervical cancer, HR-HPV, tumor-infiltrating immune cells, immunotherapy

## Abstract

The effect of cervical cancer immunotherapy is limited. Combination therapy will be a new direction for cervical cancer. Thus, it is essential to discover a novel and available predictive biomarker to stratify patients who may benefit from immunotherapy for cervical cancer. In this study, 563 participants were enrolled. Adenylate cyclase 7 (ADCY7) mRNA was detected by real-time quantitative PCR (qPCR) with cervical cytology specimens. The relationship between ADCY7 and cervical intraepithelial neoplasia in grade 2 and higher (CIN2+) was analyzed, and the optimal cut-off values of the relative expression of ADCY7 mRNA to predict CIN2+ were calculated. In addition, the clinical significance of ADCY7 in cervical cancer was determined by the Kaplan–Meier Cox regression based on the TCGA database. The mean ADCY7 mRNA expression increased significantly with cervical lesion development, especially compared with CIN2+ (*p* < 0.05). Moreover, the expression of ADCY7 increased significantly in high-risk human papillomavirus (HR-HPV) infection but not in HPV-A5/6 species. The area under the receiver operating characteristic curve (AUC) of ADCY7 was 0.897, and an optimal cut-off was 0.435. Furthermore, ADCY7 had the highest OR (OR= 8.589; 95% CI (2.281–22.339)) for detecting CIN 2+, followed by HPV genotyping, TCT, and age (OR = 4.487, OR = 2.071, and OR = 1.345; 95% CI (1.156–10.518), (0.370–8.137), and (0.171–4.694), respectively). Moreover, this study indicated that higher ADCY7 levels could be a suitable predictor for poor prognosis in cervical cancer due to immune cell infiltration. A new auxiliary predictor of CIN2+ in cervical cytology specimens is ADCY7 ≥ 0.435. Furthermore, it may be a promising prognosis predictor and potential immunotherapy target for the combined treatment of cervical cancer and possibly further block HR-HPV persistent infection.

## 1. Introduction

Cervical cancer is the fourth most common type of cancer in women [1]. According to the World Health Organization’s (WHO) Global Cancer Data Statistics Report, there are 604,127 emerging cases and 341,831 deaths annually [1]. China has 109,741 new cases and 59,060 deaths annually, which is an increasing trend [1]. Morbidity and mortality are higher in developing countries than in developed countries [2]. If no action is taken, the incidence of cervical cancer worldwide is expected to increase by at least 25 percent by 2030, to more than 700,000 cases per year [3]. Moreover, the age of onset of cervical cancer is younger than before [4]. It has become a significant challenge in female health. Therefore, more effective measures are urgently needed to intervene.

Persistent infection with high-risk human papillomavirus (HR-HPV) is a high-risk factor for cervical precancerous lesions and cervical cancer [5,6]. HPV vaccination, cervical cancer screening, and treatment of precancerous lesions are essential measures for early detection and timely treatment of early cervical cancer. HPV vaccination is a vital means to prevent cervical cancer, and HPV vaccination can reduce incidence by at least 80% [7]. Furthermore, through multi-center studies, Giorgio Bogani et al. [8] found that HPV vaccination can effectively reduce the risk of recurrence after cervical conization. However, only about 3% of school-age girls worldwide have been vaccinated against HPV, which shows that the HPV vaccination rate is not optimistic [9]. It will take a lot of time to effectively reduce the incidence of cervical cancer through mass vaccination, and many new HPV-infected people and cervical cancer patients will be diagnosed during this time. What is more, recent studies [10] have shown that HR HPV infection increases the risk of recurrence after treatment in high-grade cervical dysplasia. Traditional surgery, radiotherapy, and chemotherapy have limited efficacy in treating cervical cancer, especially for advanced and recurrent cervical cancers. Survival rates for cervical cancer patients have not improved significantly in the last 50 years [11]. Therefore, it is urgent to find new and more effective treatments.

Immunotherapy is an emerging treatment option that might be a novel option to improve the prognosis of these patients. It has recently achieved great preclinical and clinical success through immune checkpoint inhibitors (ICIs) [12,13,14]. In the clinical trial KEYNOTE-58, pembrolizumab (programmed cell death-1(PD-1) monoclonal antibody) was used to treat advanced cervical cancer, and the overall response rate (ORR) was 12.2% [15]. There are many clinical trials and preclinical studies on immunotherapy in cervical cancer. However, only pembrolizumab is currently approved by the FDA for metastatic or recurrent cervical cancer treatment. Unfortunately, ICI monotherapy has a low response rate in cervical cancer [16]. Immunotherapy provides a new direction for cervical cancer treatment. Whether monotherapy with therapeutic vaccines and ICIs may not achieve satisfactory efficacy, combination therapy is becoming the focus of many clinical trials [17]. It will be a new direction for adjuvant treatment for cervical cancer. Thus, it is essential to discover a novel target and available predictive biomarker to stratify patients who may benefit from immunotherapy for cervical cancer. Due to the limited effect of immunotherapy on cervical cancer, combinations of immunotherapies are being explored. To stratify patients who may benefit from combination therapies, predictive biomarkers for treatment outcomes will be needed.

Recently, some research [18] pointed out that adenylate cyclase 7 (ADCY7) is abnormally expressed in a variety of human cancers and is associated with mismatch repair (MMR) gene expression and DNA methyltransferase (DNMT) expression. In addition, ADCY7 expression is closely related to immune cell infiltration and immune checkpoint gene (ICG) expression. ADCY7, a member of the adenylate cyclase family, encodes a membrane protein. It is involved in extracellular signaling into intracellular reactions [19]. ADCY7 is critical in nervous system diseases, inflammatory responses, and immune responses [20,21,22]. Studies have shown that ADCY7 expression is significantly negatively correlated with overall survival in patients with acute myeloid leukemia [23] and that ADCY7 can promote the progression of acute myeloid leukemia by promoting tumor cell proliferation and migration [24]. Moreover, studies have reported that external stimuli bind to their receptors and then pass to G proteins, which can activate ADCY7. This promotes the adenosine triphosphate (ATP) conversion into cyclic adenosine-phosphate (c AMP), which acts as a second messenger to transmit signals into the cell [25,26]. In addition, the adenylate cyclase catalytic product cAMP upregulates the inhibitory molecule CTLA-4 on the surface of CD4+ T cells, which enables tumor cells to evade immune surveillance by T cells and continue to develop [27]. These findings confirm that ADCY7 is a critical molecule in tumor immune regulation. However, there are few studies about ADCY7 expression status and its biological function in cervical cancer. 

This present study aimed to evaluate the predictive value of ADCY7 expression in cervical cancer. We investigated whether the expression of ADCY7 could be a valuable biomarker to predict CIN2+ with cervical cytological specimens collected non-invasively. Additionally, this study demonstrated the prognosis of ADCY7. The results provide novel insights into the active role of ADCY7 in HPV infection and cervical cancer, thereby highlighting a potential mechanistic basis whereby ADCY7 influences immune cell interactions with tumors.

## 2. Materials and Methods

### 2.1. Patients and Study Design

We included 563 participants from the Fujian Maternity and Child Health Hospital College of Clinical Medicine for Obstetrics and Gynecology and Pediatrics, Fujian Medical University from December 2018 to May 2022. The populations had to reach the following criteria: (1) age above 20 years; (2) sexually active; (3) no history of cervical cancer, CIN, or HIV infection; and 4) did not undergo a hysterectomy or cervical surgery. The Hospital Ethics Committee approved the research of Fujian Provincial Maternity and Children’s Health Hospital, an affiliated hospital of Fujian Medical University (2022KYLLR03050), and all individuals participating in this study signed written informed consents.

### 2.2. Specimen Collection and Management

Exfoliated cervical cells were collected from cervical canals using a cytobrush. The samples were collected in vials containing preservation solutions for HPV DNA testing or in bottles of ThinPrep^®^ PreservCyt^®^ solution (Hologic, Waltham, MA, USA) for cytology examination. The storage conditions of the samples are detailed in our previous study [28,29].

### 2.3. Liquid-Based Cytology, HPV Genotyping Test 

Cytological samples were blindly examined, independent of the results from the other assays, by two experienced cytopathologists. The results were reported according to the Bethesda 2001 system [30]. If the diagnosis was inconsistent, the cervical specimens were re-analyzed, and a consensus diagnosis was reached. The PCR-RDB HPV genotyping kit (Yaneng Limited Corporation, Shenzhen, China) can discover 18 HR-HPV types and 5 LR-HPV types. All procedures were conducted according to the manufacturer’s instructions [31].

### 2.4. Histology 

The colposcopy and needle biopsy in women who were HPV-16/18 positive, with or without abnormal cytology (a grade higher than atypical squamous cells of undetermined significance (ASC-US)), were performed according to the guidelines. Women with a punch biopsy diagnosis of more than high-grade squamous intraepithelial lesions (HSIL) underwent conization by the cold knife or loop electrosurgical excision procedure cone biopsy (LEEP). Specimens were fixed in 10% formalin and routinely processed for paraffin embedding. Then, according to standard methods, 4 µm thick tissue sections were cut and stained with hematoxylin and eosin.

### 2.5. RT-PCR and Analysis of ADCY7

RT-qPCR was carried out according to the protocol. ADCY7 PCR was used with primer pairs 5′-GAT GTA CGT CGA GTG TCT CCT-3′ and 5′-CTT TGT CCA TGC GTC GAA CA-3′. GAPDH was used as an experimental control for sample quality and adequacy during the PCR process. GAPDH PCR was applied with primer pairs 5′-GGT GTG AAC CAT GAG AAG TAT GA-3′ and 5′-GAG TCC TTC CAC GAT ACC AAA G-3′. Relative levels of ADCY7 mRNA were quantified by qPCR and calculated by the 2−ΔΔCT method. 

### 2.6. Prognosis Analysis

We analyzed the OS, DSS, and PFI according to different characteristics to individualize the prognosis prediction in cervical cancer patients. All clinicopathological data were acquired from TCGA-CESC datasets, including 306 cervical squamous cell carcinoma and endocervical adenocarcinoma (CESC) sample tumor tissues and 13 normal tissues [31].

### 2.7. TIMER Database Analysis 

The Tumor Immune Estimation Resource (TIMER2.0) (https://timer.cistrome.org, accessed on 20 January 2023) is a database used for the analysis of tumor-infiltrating immune cells and various gene expression levels in different types of cancer [32]. We assessed the relationship between ADCY7 and TILs via gene modules. In addition, the relationship between ADCY7 mRNA expression and gene markers of TILs has been investigated through gene correlation research in the exploration module [33]. The statistical significance of the estimation and correlation of Spearman was analyzed by the correlation module. 

### 2.8. Statistical Analysis 

ADCY7 expression was calculated in unpaired samples using the Wilcoxon rank-sum test, while paired samples were analyzed using the Wilcoxon signed-rank test. Cox regression analyses and Kaplan–Meier analyses were performed to assess the prognostic factors. We conducted an ROC analysis using the pROC package to determine whether ADCY7 could accurately distinguish cervical intraepithelial neoplasia grade two or more (CIN2+) from healthy specimens. Furthermore, we compared the impact of ADCY7 levels on the incidence of CIN2+ using a multivariate Cox analysis. Different risks were estimated using odds ratios (ORs) and 95% confidence intervals (CIs). In all statistical analyses, *p*-values below 0.05 were considered statistically significant.

## 3. Results

### 3.1. Clinical Characteristics and Associations between Cervical Lesions and Clinicopathological Factors

The analysis of the 563 cervical tissue samples indicated that 143 (25.400%) were standard, 116 (20.604%) were classified as CIN1, 202 (35.879%) were categorized as CIN2/3, and 102 (18.117%) were diagnosed as cancer. As shown in Table 1, age, cytology, and HPV genotyping infection were significantly associated with cervical lesions. This demonstrates that abnormal cytology and older age promote the development of CIN2+. Furthermore, with the development of cervical lesions, ADCY7 mRNA levels increased, especially compared with CIN2+. This suggests that ADCY7 mRNA levels could predict the development of CIN2+ lesions (*p* < 0.001). 

The relationship between ADCY7 expression and cervical lesions and cytology is summarized in Figure 1A,B. We discovered a significant difference in means of ADCY7 expression in four groups of cervical lesions (*p* < 0.001, ANOVA test) (Figure 1A). Moreover, ADCY7 expression increased significantly in the ASCUS group and higher ASCUS group. Importantly, we found the expression of ADCY7 among different HPV genotyping. This revealed that the expression of ADCY7 increased significantly in HR-HPV positive, HPV A7 species, HPV A9 species, HPV-16/18 positive, HPV-16 positive, and HPV-18 positive samples. However, no significant difference between HPV A5/6 species positive groups was found.

ADCY7 expression is a valuable predictor for the detection of ≥CIN2. 

To predict CIN2+ lesions, ROC analyses of ADCY7 mRNA levels were performed. ADCY7 mRNA levels were evaluated individually (Figure 2). We observed that the AUC for ADCY7 mRNA levels was 0.898. Moreover, an optimal cut-off of ADCY7 mRNA levels was 0.435. As shown in Table 2, the independent factors associated with the diagnosis of CIN2+ were calculated in the multiple logistic regressions. It was demonstrated that the high expression for ADCY7 mRNA levels had the highest OR (OR= 8.589; 95% CI (2.281–22.339)), followed by HPV genotyping (OR = 4.487; 95% CI (1.156–10.518)), TCT (OR = 2.071; 95% CI (0.370–8.137)), and age (OR = 1.345; 95% CI (0.171–4.694)).

Higher ADCY7 mRNA levels predict a poor prognosis in cervical cancer. 

Based on the TCGA-CESC data sets, a Kaplan–Meier survival analysis was conducted to investigate the role of ADCY7 mRNA levels in cervical cancer survival. We evaluated the impact of ADCY7 mRNA levels on the prognosis. As shown in Figure 3, higher ADCY7 mRNA levels were significantly related to poor overall survival (OS) (HR = 2.43, 95% CI = 1.46–4.04, *p* = 0.001, Figure 3A), disease-specific survival (DSS) (HR = 2.29, 95% CI = 1.29–4.05, *p* = 0.005, Figure 3B), and the progression-free interval (PFI) (HR =1.61, 95% CI = 1.01–2.51, *p* = 0.044, Figure 3C). The univariate analysis demonstrated that the clinical stage, histological types, T stage, M stage, histologic grades, and radiation therapy affected CESC prognosis (all *p* < 0.05) (Figure 3D–T). According to the multivariate Cox regression, higher ADCY7 mRNA levels were an independent risk factor for poor OS (HR = 3.746, 95% CI = 0.588–50.170, *p* = 0.031, Figure 4A), DSS (HR = 3.039, 95% CI = 0.273–3.950, *p* = 0.014, Figure 4B), and PFI (HR =3.413, 95% CI = 0.112–8.369, *p* = 0.041, Figure 4C).

### 3.2. Association between Immune Infiltration and ADCY7 mRNA Levels in Cervical Cancer

Immune infiltration is a crucial factor related to tumor progression. In cervical cancer, ADCY7 mRNA levels are associated with immune cell infiltration levels using TIMER platforms. Figure 5 shows a strong correlation between ADCY7 mRNA levels and TIL abundance. For instance, ADCY7 mRNA levels were strongly correlated with the infiltrating degree of CD4+ T cells (rho = 0.342), T cells gamma delta (rho = 0.323), macrophages (rho = 0.325), neutrophils (rho = 0.316), M1 macrophages (rho = 0.312), myeloid dendritic cells (rho = 0.293), mast cells (rho = 0.268), endothelial cell (rho = 0.242), cancer-association fibroblasts (rho = 0.236), and NK cells (rho = 0.212). All the *p*-values were below 0.001. These results demonstrate that ADCY7 mRNA levels play an essential role in the immune infiltration in cervical cancer. In addition, this research revealed that ADCY7 mRNA levels were significantly connected with immune inhibitors, as shown in Figure 6 and Table 3. ADCY7 mRNA levels were significantly associated with immune inhibitors, such as PDCD1LG2 (rho = 0.327), CTLA4 (rho = 0.132), TIGIT (rho = 0.225), TNFRSF25 (rho = 0.228), CD40 (rho = 0.123), LAIR1 (rho = 0.256), ICOS (rho = 0.246), TNFSF4 (rho = 0.156), and TNFRSF14 (rho = −0.131). In addition, ADCY7 mRNA levels were significantly negatively associated with TNFRSF14 (rho = −0.131) and LGALS9 (rho = −0.256).

## 4. Discussion

This study demonstrated that cervical specimens with an over-expression of ADCY7 mRNA were significantly associated with CIN2+ lesions. Thus, the present study may provide information on a novel and valuable biomarker to predict the development of CIN2+ with non-invasive methods. Interestingly, this study demonstrated that a high expression of ADCY7 suggested HR-HPV infection. In this study, ADCY7 mRNA levels increased significantly with the development of the cervical lesion. Moreover, ADCY7 could be a valuable biomarker to predict CIN2+. The AUC for ADCY7 mRNA levels was 0.898. Moreover, an optimal cut-off for ADCY7 mRNA levels was 0.435. Additionally it was demonstrated that higher ADCY7 mRNA levels (≥0.435) had the highest OR (OR= 8.589; 95% CI (2.281–22.339)). Thus, this suggests that ADCY7 mRNA levels in cervical specimens can be a valuable biomarker to predict CIN2+ with non-invasive procedures. Our results may offer a helpful basis for the accurate cytological diagnosis of cervical pre-malignancy and cancer in future transitional research.

Furthermore, we also revealed that the expression of ADCY7 mRNA levels increased significantly in HR-HPV positive, HPV A7 species, HPV A9 species, HPV-16/18 positive, HPV-16 positive, and HPV-18 positive samples, but not in the HPV A5/6 species. We divided the high-risk HPV types into three groups: the HPVA5/6 species contains HPV 51, 56, and 66; the HPVA7 species contains probes for HPV 18, 39, 45, 59, and 68; and the HPVA9 species contains probes for HPV 16, 31, 33, 35, 52, and 58. It has been reported that persistent infection with high-risk human papillomavirus (HPV) is a high-risk factor for cervical precancerous lesions and cervical cancer [5,6], and HPV16 /18 accounts for most of the HPV-positive cases [34]. This study suggests that ADCY7 mRNA levels may be an auxiliary indicator for HR-HPV infection, except for HPV-A5/6 species. 

Nearly 95% of cervical cancer patients have a high risk for HPV infection [35]. However, it is not expressed in healthy tissues, so it provides a specific therapeutic target for immunotherapy of cervical cancer. T-cell receptor-engineered T cell (TCR-T) therapy, a kind of immunotherapy method, uses tumor-specific antigens as the target to obtain TCR sequences that recognize tumor antigens and then introduces the TCR sequence into the patient’s T cells through genetic engineering technology. Thus, obtained TCR-T cells can specifically recognize and kill specific tumors after in vitro expansion [36]. In addition, TCR cells also resist immune evasion by tumors [37]. A completed Phase I/II clinical trial investigating the use of E6 TCR-T cells in HPV-16-associated tumors showed that E6 TCR-T cell therapy could lead to the regression of HPV-associated epithelial cancer cells [38]. TCR-T therapy is effective in hematologic malignancies, but there is currently insufficient evidence to confirm the efficacy of TCR-T therapy in solid tumors and HPV-associated malignancies [39]. Several preclinical studies of HPV-associated tumors have shown that combining anti-PD-1 antibodies with therapeutic HPV16 E6/E7 vaccines significantly increases tumor clearance and more CD8+ T cells [40,41]. With a better understanding of cervical cancer antigen-specific T cells and immunosuppressive reversal mechanisms in the context of the tumor microenvironment (TME), it will be possible to discover more effective and safer immunotherapies for cervical cancer. Combination therapy with ICIs and targeting HPV-associated antigens will be promising. 

ADCY7 encodes a membrane protein. It is involved in extracellular signaling into intracellular reactions [19]. It is reported that external stimuli bind to their receptors and then pass to G proteins, which can activate ADCY7. This promotes the conversion of adenosine triphosphate (ATP) into cyclic adenosine-phosphate (cAMP), which acts as a second messenger to transmit signals into the cell, which in turn undergoes a series of biochemical cascades [25,26]. In addition, the adenylate cyclase catalytic product cAMP upregulates the inhibitory molecule CTLA-4 on the surface of CD4+ T cells, which enables tumor cells to evade immune surveillance by T cells and continue to develop [27]. These findings confirm that ADCY7 is a critical molecule in tumor immune regulation.

Our study revealed that higher ADCY7 mRNA levels were significantly related to poor OS (HR = 2.43, 95% CI = 1.46–4.04, *p* = 0.001), DSS (HR = 2.29, 95% CI = 1.29–4.05, *p* = 0.005), and PFI. Similarly, Giorgio Bogani. et al. [42] pointed out that HPV persistence is the only risk factor for recurrence after treatment of cervical lesions. This study further demonstrated that ADCY7 mRNA levels are significantly associated with HPV infection. Thus, we have further demonstrated that the ADCY7 expression level can be used as a valuable biomarker for HPV infection and as a prognostic indicator. Moreover, this study analyzed the correlation of ADCY7 with immune cell infiltration. We found that ADCY7 mRNA levels were strongly correlated with the infiltrating degree of CD4+ T cells, T cells gamma delta, macrophages, neutrophils, M1 macrophages, myeloid dendritic cells, mast cells, endothelial cells, cancer-association fibroblasts, and NK cells. These results demonstrate that ADCY7 mRNA levels play an essential role in the immune infiltration of cervical cancer. It is well known now that CD8 T cells play a central role in mediating anti-tumor immunity. Their effector CTLs eliminate tumor cells by recognizing tumor-associated antigens presented on primary histocompatibility complex class I (MHCI) by their expressed T cell receptor (TCR). Studies have reported that the infiltration of T cells, especially CD4+ T cells and CD8+ T cells, into the tumor microenvironment (TME) demonstrate a good prognosis in cancers, such as breast, lung, melanoma, colorectal, brain, and cervical cancer [43,44,45,46]. Conversely, suppressive immune cells, such as Treg cells, indicate a poor prognosis. Higher percentages of Tregs were significantly associated with a poor clinical outcome for patients with cervical carcinoma [45,46].

Insufficient T cell priming likely contributes to cold tumors (no T cell infiltration in TME) and unresponsiveness to immune checkpoint blockade (ICB) therapy [47]. In addition, this research revealed that ADCY7 mRNA levels were significantly associated with immune inhibitors, such as PDCD1LG2, CTLA4, TIGIT, TNFRSF14, TNFRSF25, CD40, LAIR1, LGALS9, ADORA2A, ICOS, CD276, and TNFSF4. This evidence suggests that targeting ADCY7 may enhance the anti-tumor effect by increasing the infiltration of anti-tumor immune cells and promoting the expression of the immune checkpoint. Therefore, this study further demonstrates that ADCY7 can be used as a new target for cervical cancer treatment. Moreover, it provides a new research direction for the combined treatment of cervical cancer with immunotherapy. 

If treatments for persistent HPV infection are developed, persistent HPV infection will become less frequent, significantly reducing the occurrence of high-grade intraepithelial neoplasia and cervical cancer in the cervix. Therefore, immunotherapy against HPV may have important clinical implications for preventing persistent HPV infection. This study indicates that targeting ADCY7 can serve as a new direction for immunotherapy for cervical cancer and further block HR-HPV persistent infections. This study fills the gaps in the primary and clinical research of the ADCY7 protein in the field of cervical cancer; provides theoretical support for the prevention, diagnosis, and treatment of cervical cancer; and seeks new targets for the treatment of cervical cancer. It provides valuable clinical data for ADCY7 in human biological function research, with good innovation and progress. However, there are some limitations, as follows: Firstly, we did not verify the relationship between ADCY7 expression levels and the prognosis of cervical cancer based on our clinical data in this study, which is worth further study. Secondly, the specimen size was not big enough. Therefore, increasing the sample size may be necessary in the future. 

## 5. Conclusions

This study revealed that ADCY7 expression levels significantly increased in cervical cytology and is strongly related to CIN2+ lesions. ADCY7 expression levels can be a suitable biomarker to predict CIN2+. Moreover, higher ADCY7 expression levels predict a poor prognosis in cervical cancer due to the promotion of the expression of the immune checkpoint. In addition, ADCY7 mRNA levels may be an auxiliary indicator for HR-HPV infection. These results may provide a valuable research direction for the combined treatment of cervical cancer with immunotherapy and possibly further block HR-HPV persistent infection.

## Figures and Tables

**Figure 1 biomedicines-11-00868-f001:**
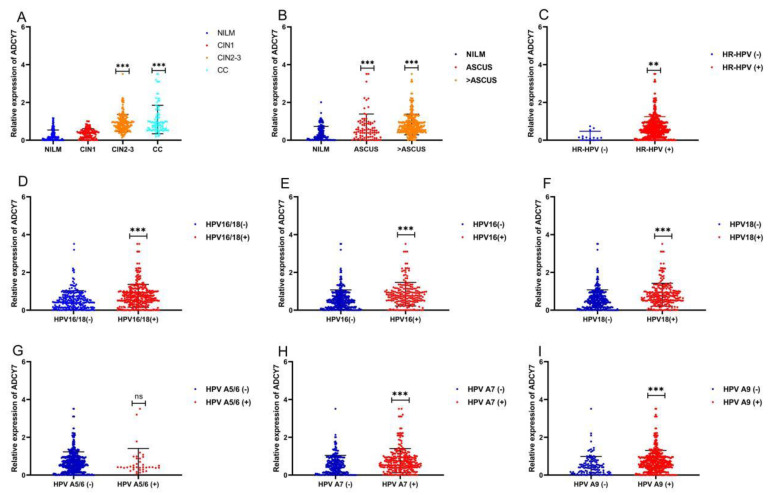
**Expression patterns of ADCY7 mRNA in cervical lesions.** (**A**) ADCY7 expression in different cervical lesions. (**B**) ADCY7 expression in different cervical cytology. (**C**–**I**) ADCY7 expression in different ADCY7 genotyping. Analysis between two groups: Wilcoxon rank-sum test; ns: *p* = 0.05 or higher; ** *p* < 0.01; *** *p* < 0.001.

**Figure 2 biomedicines-11-00868-f002:**
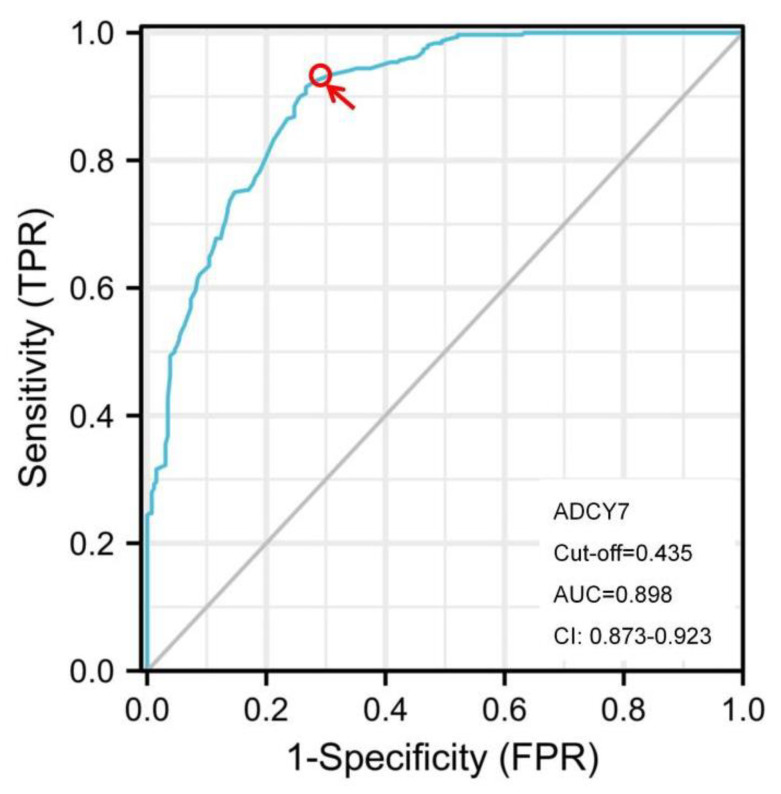
**The ROC curve analysis of ADCY7 for identifying CIN2+ disease.** Cut-off, the optimal ADCY7 expression used to predict CIN2+, was calculated according to the ROC curve. Abbreviations: CIN, cervical intraepithelial neoplasia; NILM, negative for intraepithelial lesion or malignancy; CC, cervical cancer; AUC, the area under the ROC curve; ROC, receiver operator characteristic.

**Figure 3 biomedicines-11-00868-f003:**
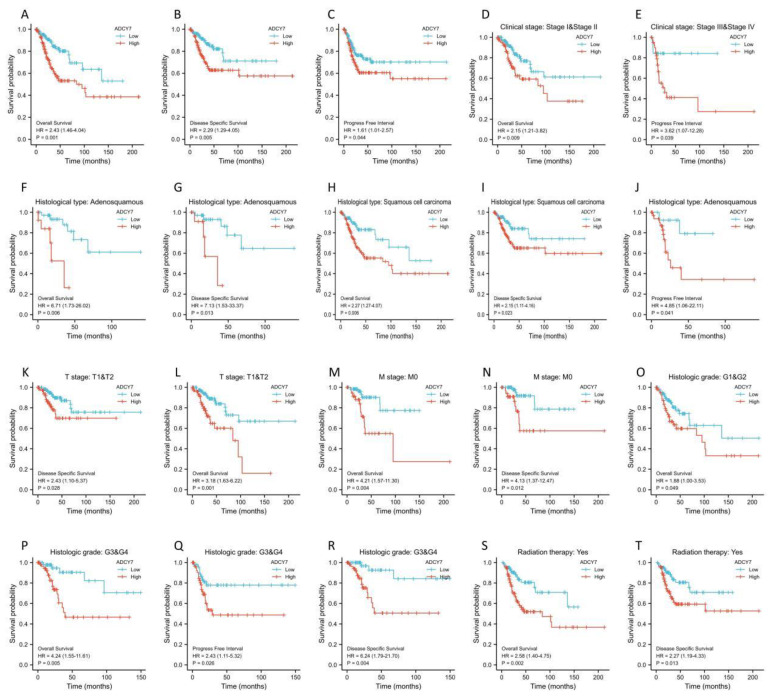
**The predictive value of ADCY7 in cervical cancer.** (**A**–**C**) Survival curves show OS, DSS, and PFI rates of CESC patients with high ADCY7 expression; (**D**–**T**) survival curves show clinical subgroup analysis of CESC patients with high ADCY7 expression.

**Figure 4 biomedicines-11-00868-f004:**
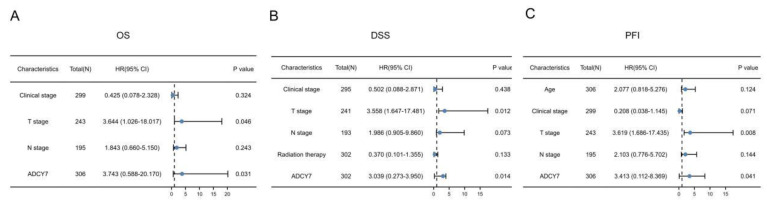
**Forest plot of the multivariate Cox regression analysis in cervical cancer.** (**A**–**C**) Risk factors with OS, DSS, and PFI rates of CESC patients with high ADCY7 expression.

**Figure 5 biomedicines-11-00868-f005:**
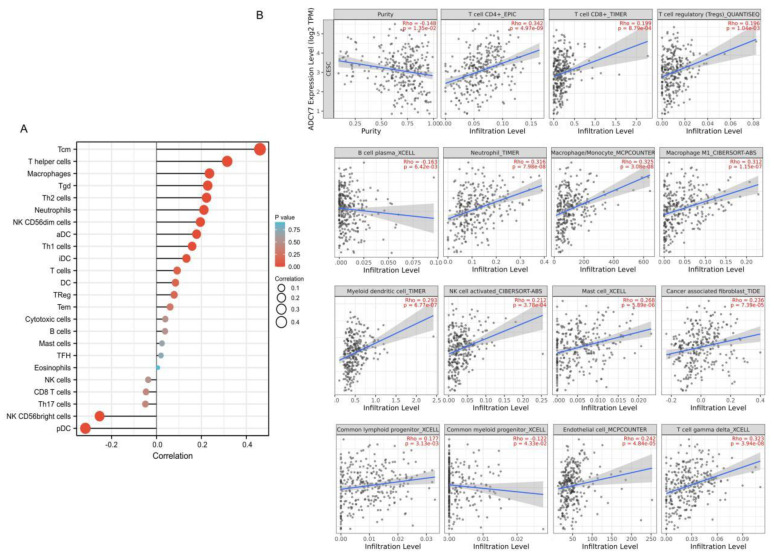
**Correlation of ADCY7 expression with immune infiltration in cervical cancer**. (**A**) Correlation between the expression of ADCY7 and the abundance of TILs in cervical cancer. (**B**) Correlation between ADCY7 expression and infiltration levels of CD4+ T cells, T cells gamma delta, macrophages, neutrophils, M1 macrophages, myeloid dendritic cells, mast cells, endothelial cells, cancer-association fibroblasts, and NK cells in cervical cancer available in the TIMER2.0 database. TILs, tumor-infiltrating lymphocytes; TIMER2.0, Tumor Immune Estimation Resource. Color images are available online.

**Figure 6 biomedicines-11-00868-f006:**
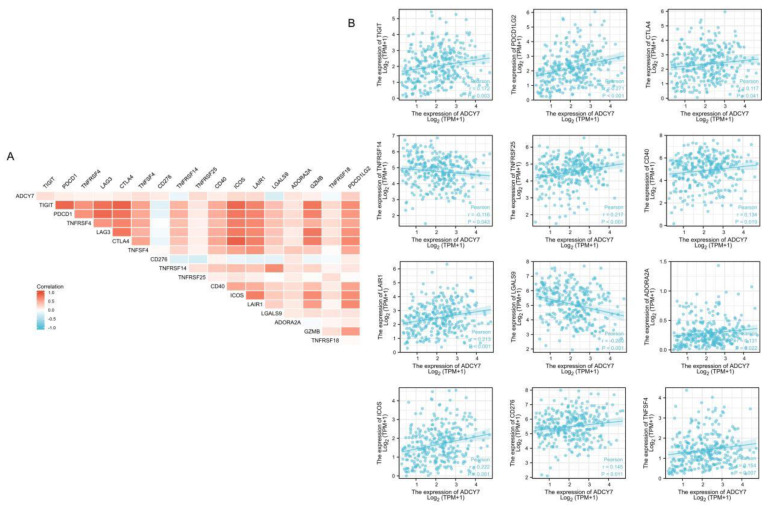
**Correlations between ADCY7 expression and various immune inhibitors in cervical cancer.** (**A**) Heat map (**B**) Scatter plo**t**, such as PDCD1LG2, CTLA4, TIGIT, TNFRSF14, TNFRSF25, CD40, LAIR1, LGALS9, ADORA2A, ICOS, CD276, and TNFSF4 in cervical cancer that are available in the TISIDB database.

**Table 1 biomedicines-11-00868-t001:** Clinical characteristics of the participants of the study.

Item	Pathology				X^2^	*p*
	Normal (n = 143)	CIN 1 (n = 116)	CIN 2–3 (n = 202)	CC (n = 102)		
Age					58.384	<0.001
<50 (N = 431)	119 (69.118%)	101 (61.062%)	162 (75.556%)	49 (53.846%)		
≥50 (N = 132)	24 (14.706%)	15 (8.850%)	40 (16.000%)	53 (43.956%)		
Cervical cytology					275.038	<0.001
NILM (N = 166)	112 (72.059%)	27 (24.778%)	20 (13.778%)	7 (10.989%)		
ASCUS (N = 87)	22 (17.647%)	31 (28.319%)	28 (14.222%)	6 (2.198%)		
>ASCUS (N = 310)	9 (10.294%)	58 (46.903%)	154 (72.000%)	89 (86.813%)		
HPV genotyping						
HPV16/18 infection					65.160	<0.001
HPV16/18 (−)	76 (50.735%)	75 (66.372%)	70 (36.000%)	16 (10.989%)		
HPV16/18 (+)	67 (49.265%)	41 (33.628%)	132 (64.000%)	86 (89.011%)		
HPV16 infection					102.715	<0.001
HPV16 (−)	111 (77.206%)	93 (79.646%)	89 (45.333%)	27 (25.275%)		
HPV16 (+)	32 (22.794%)	23 (20.354%)	113 (54.667%)	75 (74.725%)		
HPV18 infection					14.196	0.003
HPV18 (−)	108 (72.794%)	99 (85.841%)	181 (90.222%)	90 (85.714%)		
HPV18 (+)	35 (27.206%)	17 (14.159%)	21 (9.778%)	12 (14.286%)		
HPV A5/6 infection					18.831	<0.001
Negative	133 (49.265%)	98 (27.434%)	196 (54.667%)	97 (78.022%)		
Positive	10 (50.735%)	18 (72.566%)	6 (45.333%)	5 (21.978%)		
HPV A7 infection						
Negative	90 (49.265%)	85 (27.434%)	177 (54.667%)	86 (78.022%)	33.455	<0.001
Positive	53 (50.735%)	31 (72.566%)	25 (45.333%)	16 (21.978%)		
HPV A9 infection					55.814	<0.001
Negative	63 (49.265%)	34 (27.434%)	23 (54.667%)	15 (78.022%)		
Positive	80 (50.735%)	82 (72.566%)	179 (45.333%)	87 (21.978%)		
ADCY7	0.246 ± 0.025	0.373 ± 0.023	0.916 ± 0.032	1.096 ± 0.075	764.228	<0.001

CIN, cervical intraepithelial neoplasia; NILM, negative for intraepithelial lesion or malignant; ASC-US, atypical squamous cells of undetermined significance; CC, cervical cancer. *p* < 0.05.

**Table 2 biomedicines-11-00868-t002:** Independent predictors for the detection of CIN2+ lesions.

	Category	OR	95% CI	*p*
Age	<50	1	Reference	0.003
	≥50	1.345	0.171–4.694	
TCT	<ASCUS	1	Reference	<0.001
	≥ASCUS	2.071	0.370–8.137	
HPV genotyping	non-16/18 (+)	1	Reference	<0.001
	16/18 (+)	4.487	1.156–10.518	
ADCY7	Low	1	Reference	0.001
	High (≥0.435)	8.589	2.281–22.339	

Note: The cut-off of PMR values was defined by the ROC curve analysis. Abbreviations: TCT, Thinprep cytologic test; ASC-US, atypical squamous cells of undetermined significance; OR, odds ratio; 95% CI, 95% confidence interval.

**Table 3 biomedicines-11-00868-t003:** Correlation analysis between ADCY7 and related genes and markers of immune cells of CESC in the Tumor Immune Estimation Resource (TIMER2.0).

Description	Gene Markers	Cor	*p*

CD8+ T cell	CD8A	0.148	*
	CD8B	−0.074	0.194
T cell (general)	CD3D	0.014	0.802
	CD3E	0.139	*
	CD2	0.111	0.152
B cell	CD19	0.097	0.090
	CD79A	0.099	0.085
Monocyte	CD86	0.216	***
	CD115 (CSF1R)	0.304	***
TAM	CCL2	0.110	0.055
	CD68	0.016	0.784
	IL10	0.265	***
M1 Macrophage	INOS (NOS2)	−0.142	*
	IRF5	0.264	***
	COX2(PTGS2)	0.095	0.098
M2 Macrophage	CD163	0.327	***
	VSIG4	0.308	***
	MS4A4A	0.214	***
	CD11b (ITGAM)	0.244	***
	CCR7	0.167	**
Natural killer cell	KIR2DL1	0.132	*
	KIR2DL3	0.160	**
	KIR2DL4	0.108	0.060
	KIR3DL1	0.154	**
	KIR3DL2	0.207	**
	KIR3DL3	0.137	*
	KIR2DS4	0.176	**
Dendritic cell	HLA-DPB1	−0.067	0.245
	HLA-DQB1	−0.066	0.252
	HLA-DRA	−0.018	0.754
	HLA-DPA1	0.027	0.641
	BDCA-1(CD1C)	0.073	0.200
	BDCA-4(NRP1)	0.267	***
	CD11c (ITGAX)	0.322	***
Th1	T -bet (TBX21)	0.187	***
	STAT4	0.207	***
	STAT1	0.351	***
	IFN--γ(IFNG)	0.139	*
	TNF-α (TNF)	0.174	**
Th2	GATA3	0.254	***
	STAT6	0.374	***
	STAT5A	0.205	***
	IL13	−0.014	0.802
Tfh	BCL6	0.036	0.528
	IL21	0.237	**
Th17	STAT3	0.387	***
	IL17A	−0.004	0.951
Treg	FOXP3	0.198	***
	CCR8	0.438	***
	STAT5B	0.328	***
	TGF-β (TGFB1)	0.467	***
T cell exhaustion	PD-1 (PDCD1)	0.096	0.093
	PDL1(PDCD1LG2)	0.271	***
	CTLA4	0.117	*
	LAG3	0.057	0.317
	TIM-3 (HAVCR2)	0.212	***
	GZMB	0.105	0.066

Note: CESC, cervical squamous cell carcinoma and endocervical adenocarcinoma; TAM, tumor-associated macrophage; Th, T helper cell; Tfh, follicular helper T cell; Treg, regulatory T cell; Cor, R-value of Spearman’s correlation; None, correlation without adjustment; Purity, correlation adjusted by purity. * *p* < 0.05; ** *p* < 0.01; *** *p* < 0.001.

## Data Availability

Not applicable.

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
