# Peer review of "ADCY7 mRNA Is a Novel Biomarker in HPV Infection and Cervical High-Grade Squamous Lesions or Higher"

_biomedicines, 2023, doi:10.3390/biomedicines11030868_

Round 1
Reviewer 1 Report
I read with great interest the Manuscript titled "ADCY7 is a Novel Biomarker with Infection and Cervical High-Grade Squamous Lesions and More" which falls within the aim of the Journal.
In my honest opinion, the topic is interesting enough to attract the readers’ attention. Methodology is accurate and conclusions are supported by the data analysis. Nevertheless, authors should clarify some point and improve the discussion citing relevant and novel key articles about the topic.
-The whole text should be corrected by a native English speaker in order to make the work clearer and more readable.
-The introduction should be extended and completed. I find interesting a reference to the efforts made for the prevention and early diagnosis of gynecological cancers (see PMID: 29958629).
- Discussions can be expanded and improved by citing relevant articles (I suggest authors to read and insert in references the following article PMID: 32893030).
Considered all this points, I think it could be of interest for the readers and, in my opinion, it deserves the priority to be published after minor revisions.
Author Response
Biomedicine
Professor Editorial Office
Editor-in-Chief
Manuscript title: ADCY7 is a Novel Biomarker with Infection and Cervical High-Grade Squamous Lesions and More
Submission ID: biomedicines-2091490
Dear editors and reviewers :
Thank you so much for your arranging a timely review for our manuscript. We are excited to receive the letter from your editorial office. We would like to thank all members of the editor team and the peer reviewers for their helpful suggestions and remarks. We would like thank you again for the chance to submit a revision version. As soon as we received the last decision letter, we held a group meeting to address all of the critiques mentioned, with particular focus on the issues that need to be improved.
To the best of our knowledge, we did consider all topics that required a further attention. Some parts of the manuscript were rewritten and marked as yellow bright. We are confident that the present version of the manuscript is far more stringent and straightforward. Furthermore, an English language editor has reviewed the revised manuscript and corrected any grammar errors. We wish to refer to the comments systematically, and detailed corrections are listed below point by point.
Response to Reviewers
Response to reviewer #1
- In my honest opinion, the topic is interesting enough to attract the readers’ attention. Methodology is accurate and conclusions are supported by the data analysis. Nevertheless, authors should clarify some point and improve the discussion citing relevant and novel key articles about the topic.The introduction should be extended and completed. I find interesting a reference to the efforts made for the prevention and early diagnosis of gynecological cancers (see PMID:29958629).
Response: First of all, thank you for your recognition of this article. We do agree with the reviewer’s suggestion. According to the reviewer’s suggestion, we have added some contents and corresponding references in the introduction in P4, line119-120.
- Discussions can be expanded and improved by citing relevant articles (I suggest authors to read and insert in references the following article PMID:32893030).
Response: Thank you for your suggestion. That is really an excellent comment. We do agree with the reviewer’s suggestion. We have added some context in the discussions in P11, line304-307.
- The whole text should be corrected by a native English speaker in order to make the work clearer and more readable.
Response: Thank you for your suggestion.We are very sorry for our poor English expression. An English language editor has reviewed the revised manuscript and corrected any grammar errors.
Finally, we thank you again for your help in improving the quality of our manuscript. We tried our best to improve our manuscripts to match your journals criterion. And all the revisions were marked as bright red in this version.
We do earnestly thank to the editors/reviewers’ work and their helpful comments and suggestions and we would like address our deeply gratitude.
Sincerely,
Correspondence to:
Yang Xiang, MD. Professor, Peking Union Medical College Hospital, Chinese Academy of Medical Sciences, Peking Union Medical College, No.1 Shuai fu yuan Wang fu jing, Dong cheng District, Beijing, 100730, People’s Republic of China.
Email: xiangy@pumch.cn.
Tel: 0086-01065296068
Fax: 0086-01065296218
Reviewer 2 Report
Authors presented a study to investigate the use of ADCY7 as biomarker to stratify patients with CIN2+ and find those who may benefit from immunotherapy for cervical cancer.
-The readability of the manuscript can be improved.
-I find interesting to include in the introduction the long-term role of adjuvant vaccination against HPV (see: 33271963)
- I recommend to add further details to discuss the outcomes women affected by high-grade cervical dysplasia (authors may refer to PMID: 33514481).
To be re-evaluated after suggested changes.
Author Response
Biomedicine
Professor Editorial Office
Editor-in-Chief
Manuscript title: ADCY7 is a Novel Biomarker with Infection and Cervical High-Grade Squamous Lesions and More
Submission ID: biomedicines-2091490
Dear editors and reviewers :
Thank you so much for your arranging a timely review for our manuscript. We are excited to receive the letter from your editorial office. We would like to thank all members of the editor team and the peer reviewers for their helpful suggestions and remarks. We would like thank you again for the chance to submit a revision version. As soon as we received the last decision letter, we held a group meeting to address all of the critiques mentioned, with particular focus on the issues that need to be improved.
To the best of our knowledge, we did consider all topics that required a further attention. Some parts of the manuscript were rewritten and marked as yellow bright. We are confident that the present version of the manuscript is far more stringent and straightforward. Furthermore, an English language editor has reviewed the revised manuscript and corrected any grammar errors. We wish to refer to the comments systematically, and detailed corrections are listed below point by point.
Response to Reviewers
Response to reviewer #2
- Authors presented a study to investigate the use ofADCY7 as biomarker to stratify patients with CIN2+ and find those who may benefit from immunotherapy for cervical cancer.The readability of the manuscript can be improved.
Response: Thanks for your comments. We are very sorry for causing you a very bad reading experience because of our poor English expression. We have carefully read and checked the whole manuscript, and revised the doubtful points in the revision.
- I find interesting to include in the introduction the long-term role of adjuvant vaccination against HPV (see:33271963)
Response: Thank you for your suggestion. We do agree with the reviewer’s suggestion. We have added this section in P4, line103-104.
- I recommend to add further details to discuss theoutcomes women affected by high-grade cervical dysplasia (authors may refer to PMID: 33514481).
Response: Thank you for your suggestion. We do agree with the reviewer’s suggestion. We have added this section in P4, line108-109.
Finally, we thank you again for your help in improving the quality of our manuscript. We tried our best to improve our manuscripts to match your journals criterion. And all the revisions were marked as bright red in this version.
We do earnestly thank to the editors/reviewers’ work and their helpful comments and suggestions and we would like address our deeply gratitude.
Sincerely,
Correspondence to:
Yang Xiang, MD. Professor, Peking Union Medical College Hospital, Chinese Academy of Medical Sciences, Peking Union Medical College, No.1 Shuai fu yuan Wang fu jing, Dong cheng District, Beijing, 100730, People’s Republic of China.
Email: xiangy@pumch.cn.
Tel: 0086-01065296068
Fax: 0086-01065296218
Reviewer 3 Report
The manuscript describes the identification of ADCY7 mRNA in cervical cytology samples as a possible diagnostic biomarker for high-grade cervical lesions (CIN2+) caused by HPV infections. Discovery of non-invasive biomarkers is of high interest as an extra tool to reduce the referral of women to colposcopy based on HPV-based cervical screening only.
Comments on overall manuscript:
The rationale to study ADCY7 mRNA expression in cervical disease is not very clear. Did the authors have any preliminary data that triggered this study?
The discussion on higher levels of tumor-infiltrating lymphocytes and poor prognosis should be rewritten as it is not clear that the TILs are most likely tumor-suppressive. It is confusing that the authors refer to studies that have shown 'that infiltration of T cells, especially CD4+ T cells and CD8+ T cells, into the tumor microenvironment (TME) demonstrated a good prognosis in cancer, such as breast, lung, melanoma, colorectal, and brain cancer [40-41]', but not to studies that have studied TILs in cervical cancer (e.g., Jordanova et al., Clin Cancer Res 2008, 14:2028-35 or Shah et al., Cell Mol Immunol 2011, 8:59-66), where they are prognostic for poor outcomes. The 'increased infiltration of anti-tumor immune cells' that is mentioned in the conclusion would normally be a positive effect, that should lead to tumor regression.
The manuscript needs extensive editing of the English language throughout the document. Here are a few examples from the title and abstract:
Title:
· It should be clarified that the level of ADCY7 mRNA is a biomarker for high-grade cervical lesions.
· If ‘infection’ is mentioned in the title, it should include ‘HPV’ (HPV infection).
· It is not clear what ‘and more’ means.
Abstract:
· What kind of combination therapy is meant here?
· What does an ‘available’ biomarker mean?
· The authors mention that ‘The relationship between ADCY7 and cervical intraepithelial neoplasia in two grades and more (CIN2+) was analyzed’, but they analyzed the relationship between ADCY7 (mRNA expression) from normal cytology to high-grade cervical disease.
· It should be clarified which database was used in the sentence ‘In addition, the clinical significance of ADCY7 in cervical cancer was determined by Kaplan-Meier Cox regression based on the database.’
· The sentence ‘The mean ADCY7 increased significantly with cervical lesion development, especially compared with CIN2+ (p<0.05)’ should be corrected. For example to: The mean ADCY7 expression increased with cervical lesion development, with a statistical significant difference for CIN2+ compared to lower grade lesions.
· ‘Moreover, the expression of ADCY7 increased significantly in high-risk human papillomavirus (HR-HPV) infections but not in infections with HPV-A5/6 species.’
· It is unusual to start sentences with ‘And…’.
· In the sentence ‘Furthermore, the ADCY7 had the highest OR (OR= 8.589; 95% CI (2.281-22.339)) for detecting CIN 2+.’ odds ratio should be written in full and it should be clarified which other factors (e.g., HPV16/18 status, age) were analyzed.
Comment on use of TIMER methodology:
The authors should clarify how they generated the data in Table 3 in more detail. While the numbers for the correlation between ADCY7 expression and different TILs can be easily verified, this is not directly obvious for the data in Table 3. For example, the gene correlation between ADCY7 and CD8A in the CESC dataset seems to be 0.172 (with purity assessment) on the TIMER2.0 website, while the authors show 0.148. Furthermore, a reference or explanation is missing why the authors selected the indicated gene markers for the various cell types (such as CD8A, CD8B and CD3D for CD8+ T cells).
Small comment: the correlations between ADCY7 and TNFSR14 or LGALS9 should be indicated as negatively correlated, rho= -0.116 or rho= -0.260, respectively.
Please, clarify the apparent contradiction between 'This study demonstrated the prognosis of ADCY7.' in the introduction, 'We evaluated the impact of ADCY7 mRNA levels on the prognosis' in the results section and 'we did not use our data to verify the relationship between ADCY7 expression levels and the prognosis of cervical cancer in this study.' in the discussion.
Comments on references
The statement ‘Cervical cancer (CC) is the most common malignancy in women [1].’ is incorrect and Reference 1 does not mention this. Cervical cancer is the 4th most common type of cancer in women: https://www.wcrf.org/cancer-trends/worldwide-cancer-data/
2 out of the first 3 references are incorrect: for reference 1 see above and reference 3 does not mention cervical cancer rates in China (it is not even about cervical cancer). Check all references.
Formatting:
Please, check formatting of all references (e.g., abbreviations of journal names, year between brackets or not, titles included or not, spaces between names and initials, inclusion of 'doi' or not, etc.)
Please, check complete document for proper use of spaces.
Author Response
Biomedicine
Professor Editorial Office
Editor-in-Chief
Manuscript title: ADCY7 is a Novel Biomarker with Infection and Cervical High-Grade Squamous Lesions and More
Submission ID: biomedicines-2091490
Dear editors and reviewers :
Thank you so much for your arranging a timely review for our manuscript. We are excited to receive the letter from your editorial office. We would like to thank all members of the editor team and the peer reviewers for their helpful suggestions and remarks. We would like thank you again for the chance to submit a revision version. As soon as we received the last decision letter, we held a group meeting to address all of the critiques mentioned, with particular focus on the issues that need to be improved.
To the best of our knowledge, we did consider all topics that required a further attention. Some parts of the manuscript were rewritten and marked as yellow bright. We are confident that the present version of the manuscript is far more stringent and straightforward. Furthermore, an English language editor has reviewed the revised manuscript and corrected any grammar errors. We wish to refer to the comments systematically, and detailed corrections are listed below point by point.
Response to Reviewer #3
- The rationale to study ADCY7 mRNA expression in cervical disease is not very clear. Did the authors have any preliminary data that triggered this study?
Response: Thank you for your suggestion.That is really an excellent comment. Recently, some research pointed out that ADCY7 is abnormally expressed in a variety of human cancers and is associated with mismatch repair (MMR) gene expression, and DNA methyltransferase (DNMT) expression. In addition, ADCY7 expression is closely related to immune cell infiltration and immune checkpoint gene (ICG) expression. It suggested that ADCY7 may be a prognostic biomarker for tumorigenesis (PMID: 34539183). However, we found there is no study analyzed the association between ADCY7 and cervical cancer. So we wanted to investigate the relationship between ADCY7 and cervical cancer. We have added relevant content has been added in the introduction in P5, line127-130.
- The discussion on higher levels of tumor-infiltrating lymphocytes and poor prognosis should be rewritten as it is not clear that the TILs are most likely tumor-suppressive.
Response: Thank you for your suggestion. We do agree with the reviewer’s suggestion. But we did not find some sentence about "higher levels of tumor-infiltrating lymphocytes and poor prognosis". We carefully checked the whole manuscript and made corrections to confirm there are no same points.
- It is confusing that the authors refer to studies that have shown 'that infiltration of T cells, especially CD4+ T cells and CD8+ T cells, into the tumor microenvironment (TME) demonstrated a good prognosis in cancer, such as breast, lung, melanoma, colorectal, and brain cancer [40-41]', but not to studies that have studied TILs in cervical cancer (e.g., Jordanova et al., Clin Cancer Res 2008, 14:2028-35 or Shah et al., Cell Mol Immunol 2011, 8:59-66), where they are prognostic for poor outcomes. The 'increased infiltration of anti-tumor immune cells' that is mentioned in the conclusion would normally be a positive effect, that should lead to tumor regression.
Response: Thank you for your suggestion.That is really an excellent comment. We do agree with the reviewer’s suggestion. Firstly, we really did not discuss the prognosis of TILs in cervical cancer and thanks to the reviewers for providing two valuable references. Secondly, thank you for helping us find the vague description in the manuscript. The relationship between TILs and prognosis mentioned in the manuscript should be explained in detail. TILs contain various immune cells. The infiltration of T cells, especially CD4+ T cells and CD8+ T cells, into the tumor microenvironment (TME) demonstrated a good prognosis. Conversely, the suppressive immune cells, such as Tregs cells predict a poor prognosis. We are so sorry to cause a bad reading experience for poor English. We have carefully checked and revised the relevant content in the discussion in P11, line322-327.
- Title:(1) It should be clarified that the level of ADCY7 mRNA is a biomarker for high-grade cervical lesions; (2) If ‘infection’ is mentioned in the title, it should include ‘HPV’ (HPV infection); (3) It is not clear what ‘and more’ means.
Response: Firstly, thank you for your valuable comments.The 'and more' in the title means higher level of cervical lesions. We do agree with the reviewer’s suggestion. Therefore, we revised the title “ADCY7 mRNA is a Novel Biomarker with HPV Infection and Cervical High-Grade Squamous Lesions or Higher” in the manuscript.
- Abstract:(1) What kind of combination therapy is meant here?(2) What does an ‘available’ biomarker mean?(3) ·The authors mention that ‘The relationship between ADCY7 and cervical intraepithelial neoplasia in two grades and more (CIN2+) was analyzed’, but they analyzed the relationship between ADCY7 (mRNA expression) from normal cytology to high-grade cervical disease. (4) ·It should be clarified which database was used in the sentence ‘In addition, the clinical significance of ADCY7 in cervical cancer was determined by Kaplan-Meier Cox regression based on the database.’ (5)·‘Moreover, the expression of ADCY7 increased significantly in high-risk human papillomavirus (HR-HPV) infections but not in infections with HPV-A5/6 species.’·It is unusual to start sentences with ‘And…’.(6)·In the sentence ‘Furthermore, the ADCY7 had the highest OR (OR= 8.589; 95% CI (2.281-22.339)) for detecting CIN 2+.’ odds ratio should be written in full and it should be clarified which other factors (e.g., HPV16/18 status, age) were analyzed.
Response: Thank you for your suggestion. (1) The combination therapy means two immune checkpoint inhibitors combination, or single immune checkpoint inhibitor combination chemotherapy. (2) The 'available' means that the sensitivity and specificity are at a good level, the detection method simple, and the specimen source convenient. (3) Indeed, we analyzed the relationship between ADCY7 (mRNA expression) from normal cytology to high-grade cervical disease. The normal group was used as a control to verify the difference in ADCY7 expression levels between high-grade cervical disease and the normal group. (4) Yes, we should wrote clearly the specific database (TCGA database), and we have revised the relevant content in P3, line69. (5) Ok, the ‘And…’has been removed in the revision. (6) We do agree with the reviewer’s suggestion. We have added the context in P4, line74-76.
- Comment on use of TIMER methodology:The authors should clarify how they generated the data in Table 3 in more detail. While the numbers for the correlation between ADCY7 expression and different TILs can be easily verified, this is not directly obvious for the data in Table 3. For example, the gene correlation between ADCY7 and CD8A in the CESC dataset seems to be 0.172 (with purity assessment) on the TIMER2.0 website, while the authors show 0.148. Furthermore, a reference or explanation is missing why the authors selected the indicated gene markers for the various cell types (such as CD8A, CD8B and CD3D for CD8+ T cells).
Response: Thank you for your suggestion.That is really an excellent comment. We do agree with the reviewer’s suggestion. Firstly, we further described the method how to obtain the data in Table 3 in P7, line197-198. Secondly, we checked all of the Gene marker's data and found that some of the data was different. It may be attribute to the data updates from online sites. We have checked and revised all data in the Table 3 and in P9, line257-262. with the latest results. Finally, these genes are common gene markers for these immune cells. In addition, Chen L. et al. also reported these genes to label different immune cells, and we have added corresponding references.
- Small comment: the correlations between ADCY7 and TNFSR14 or LGALS9 should be indicated as negatively correlated, rho= -0.116 or rho= -0.260, respectively.
Response: Thank you for your suggestion. We do agree with the reviewer’s suggestion. We have added the sentence in P9, line260-262.
- Please, clarify the apparent contradiction between 'This study demonstrated the prognosis of ADCY7.' in the introduction, 'We evaluated the impact of ADCY7 mRNA levels on the prognosis' in the results section and 'we did not use our data to verify the relationship between ADCY7 expression levels and the prognosis of cervical cancer in this study.' in the discussion.
Response: Thank you for your professional suggestion.We are very sorry for our incorrect writing. What we wanted to express is that we analyzed the prognostic effect of ADCY7 mRNA levels by comparing patient clinical information in public databases. But we did not use our own clinical data to further verify the prognostic effect of ADCY7 mRNA levels. We have revised the relevant sentences in P12, line346-347.
- Comments on references:The statement ‘Cervical cancer (CC) is the most common malignancy in women [1].’ is incorrect and Reference 1 does not mention this. Cervical cancer is the 4th most common type of cancer in women: https://www.wcrf.org/cancer-trends/worldwide-cancer-data/ï¼›out of the first 3 references are incorrect: for reference 1 see above and reference 3 does not mention cervical cancer rates in China (it is not even about cervical cancer). Check all references.
Response: Thank you for your suggestion again. According to the opinions of the reviewers, we carefully checked all references. We apologize for errors in the citation references. We have revised and checked all references.
- The manuscript needsextensive editing of the English language throughout the document. Here are a few examples from the title and abstract:
Response: Thank you for your suggestion.We are very sorry for our poor English expression. An English language editor has reviewed the revised manuscript and corrected any grammar errors.
Finally, we thank you again for your help in improving the quality of our manuscript. We tried our best to improve our manuscripts to match your journals criterion. And all the revisions were marked as bright red in this version.
We do earnestly thank to the editors/reviewers’ work and their helpful comments and suggestions and we would like address our deeply gratitude.
Sincerely,
Correspondence to:
Yang Xiang, MD. Professor, Peking Union Medical College Hospital, Chinese Academy of Medical Sciences, Peking Union Medical College, No.1 Shuai fu yuan Wang fu jing, Dong cheng District, Beijing, 100730, People’s Republic of China.
Email: xiangy@pumch.cn.
Tel: 0086-01065296068
Fax: 0086-01065296218
Round 2
Reviewer 3 Report
Thanks for addressing my comments and questions.
Please, check reference 4 as it does not address the age of onset of cervical cancer
Please, check reference 7 as it is not clear where the number of 80% is derived from.
Please, clarify which household gene was used in the 2−ΔΔCT method.
Some suggestions for improvements, but the manuscript could be improved in many places:
Suggestion for Background: ‘Due to the limited effect of immunotherapy on cervical cancer, combinations of immunotherapies are being explored. To stratify patients who may benefit from combination therapies, predictive biomarkers for treatment outcome will be needed.’
Line 66 - Correct ‘in two grades and more’ to ‘grade 2 and higher’
Line 67 - ‘optimal cut-off values of relative expression of ADCY7 mRNA’
Line 70 – ‘The mean ADCY7 mRNA expression increased’
Line 72 – ‘It revealed that The area under the…..’
Line 74 – ‘Furthermore, the ADCY7 had the highest OR…’
Line 96 – ‘Moreover, the age of onset of cervical cancer is younger [4]’. Younger than what?
Line 102-13 – It is unclear what ‘contain’ means in the sentence: ‘HPV vaccination can contain at least 80% of the target population [7]’.
Line 222 - 'ROC analyses of ADCY7 mRNA levels was performed'
Author Response
Biomedicine
Professor Editorial Office
Editor-in-Chief
Manuscript title: ADCY7 is a Novel Biomarker with Infection and Cervical High-Grade Squamous Lesions and More
Submission ID: biomedicines-2091490
Dear editors and reviewers :
Thank you so much for your arranging a timely review for our manuscript. We are excited to receive the letter from your editorial office. We would like to thank all members of the editor team and the peer reviewers for their helpful suggestions and remarks. We would like thank you again for the chance to submit a revision version. As soon as we received the last decision letter, we held a group meeting to address all of the critiques mentioned, with particular focus on the issues that need to be improved.
To the best of our knowledge, we did consider all topics that required a further attention. Some parts of the manuscript were rewritten and marked as yellow bright. We are confident that the present version of the manuscript is far more stringent and straightforward. Furthermore, an English language editor has reviewed the revised manuscript and corrected any grammar errors. We wish to refer to the comments systematically, and detailed corrections are listed below point by point.
Response to Reviewers
Response to reviewer #3
- Please, check reference 4 as it does not address the age of onset of cervical cancer
Response: First of all, thank you for your suggestion again. According to the opinions of the reviewers, we carefully checked all references. We are so sorry for citing indirect references. We have revised the references.
- Please, check reference 7 as it is not clear where the number of 80% is derived from
Response: Thank you for your suggestion. We do agree with the reviewer’s suggestion.Firstly, this sentence means that HPV vaccination can reduce at least 80% incidence. Secondly, we are so sorry for citing indirect references. We have revised the the sentence and references.
- Please, clarify which household gene was used in the 2−ΔΔCT method.
Response: Thank you for your suggestion.GAPDH was the household gene in the manuscript, which has been described in the P8, line 182-183.
- (1) Suggestion for Background: ‘Due to the limited effect of immunotherapy on cervical cancer, combinations of immunotherapies are being explored. To stratify patients who may benefit from combination therapies, predictive biomarkers for treatment outcome will be needed.’Line 66 - Correct ‘in two grades and more’ to ‘grade 2 and higher’; (2) Line 67 - ‘optimal cut-off values of relative expression of ADCY7 mRNA’; (3) Line 70 – ‘The mean ADCY7 mRNA expression increased’; (4) Line 72 – ‘It revealed that The area under the…..’; (5) Line 74 – ‘Furthermore, the ADCY7 had the highest OR…;(6) Line 222 - 'ROC analyses of ADCY7 mRNA levels was performed'.
Response: Thank you for your suggestion. We do agree with the reviewer’s suggestion. We have modified the above sentences in the corresponding places in the manuscript and marked as yellow bright.
- Line 96 – ‘Moreover, the age of onset of cervical cancer is younger [4]’. Younger than what?
Response: Thank you for your professional suggestion.We are very sorry for our poor expression. This sentence means that the age of onset of cervical cancer is younger than before. We have modified this sentence in P4, line 96-97.
- Line 102-13 – It is unclear what ‘contain’ means in the sentence: ‘HPVvaccination can contain at least 80% of the target population [7]’.
Response: Thank you for helping us find the vague description in the manuscript. We
have revised this sentence in P4, line 102-103.
Finally, we thank you again for your help in improving the quality of our manuscript. We tried our best to improve our manuscripts to match your journals criterion. And all the revisions were marked as bright red in this version.
We do earnestly thank to the editors/reviewers’ work and their helpful comments and suggestions and we would like address our deeply gratitude.
Sincerely,
Correspondence to:
Yang Xiang, MD. Professor, Peking Union Medical College Hospital, Chinese Academy of Medical Sciences, Peking Union Medical College, No.1 Shuai fu yuan Wang fu jing, Dong cheng District, Beijing, 100730, People’s Republic of China.
Email: xiangy@pumch.cn.
Tel: 0086-01065296068
Fax: 0086-01065296218